

**Early Twentieth Century Southern Hemisphere Cooling**
Stefan Brönnimann,[1,2] Yuri Brugnara,[1,2,*] Clive Wilkinson[3]
*[1]Institute of Geography, University of Bern, Switzerland*
*[2]Oeschger Centre for Climate Change Research, University of Bern, Switzerland*
*[3]CSW Associates and Climatic Research Unit, University of East Anglia, Norwich, UK*
*\* now at Empa, Laboratory for Air Pollution and Environmental Technology, Dübendorf, Switzerland*
*Corresponding author: S. Brönnimann, stefan.broennimann@unibe.ch*
**Abstract**
Global surface air temperature increased by ca. 0.5 °C from the 1900s to the mid-1940s, also known as Early
Twentieth Century Warming (ETCW). However, the ETCW started from a particularly cold phase, peaking in
1908-1911. The cold phase was global but more pronounced in the Southern Hemisphere than in the Northern
Hemisphere and most pronounced in the Southern Ocean, raising the question whether uncertainties in the data
might play a role. Here we analyse this period based on reanalysis data and reconstructions, complemented with
newly digitized ship data from 1903-1916 as well as land observations. The cooling is seen consistently in
different data sets, though with some differences. Results suggest that the cooling was related to a La Niña-like
pattern in the Pacific, a cold tropical and subtropical South Atlantic, a cold extratropical South Pacific, and cool
Southern midlatitude land areas. The Southern Annular Mode was positive, with a strengthened Amundsen-
Bellingshausen seas low, although the spread of the data products is considerable. All results point to a real
climatic phenomenon as the cause of this anomaly and not a data artefact. Atmospheric model simulations are
able to reproduce temperature and pressure patterns, consistent with a real and perhaps ocean-forced signal.
Together with two volcanic eruptions just before and after the 1908-1911 period, the early 1900s provided a
cold start into the ETCW.

**1    Introduction**
Global warming since the early 20th century proceeded in two phases, the so-called Early Twentieth
Century Warming (ETCW) from ca. 1905 to 1945 (Brönnimann, 2009, see review by Hegerl et al.,
2018) followed by a plateau phase in the 1950s and 1960s, and the strong warming since 1970. The
ETCW has been a matter of keen scientific interest, but the focus was mostly on the trend or on the
peak phase in the 1940s. Interestingly, the ETCW started with a clear dip in global temperatures
around 1910, which is more pronounced in sea-surface temperature (SST) than over land and which,
if analysed spatially, is most pronounced in the Southern Hemisphere (Hegerl et al., 2018). If real, one
might expect to find anomalous atmospheric circulation along with this change.



Atmospheric circulation during the first decade of the 20[th] century has not received much attention, particularly not in the Southern Hemisphere. The Southern Oscillation Index (SOI) shows a tendency towards a strengthened Walker circulation (Cane, 2005). Reconstructions of the Southern Annular Mode (SAM) (e.g., Abram et al., 2014) indicate neutral values. The most comprehensive analysis was performed by Connolly (2020) and Fogt and Connolly (2021). They presented a new reconstruction of pressure back to 1905 and found that the SAM signal also dominated in the early 20[th] century, however, without addressing specifically this period. They also found considerable differences between station-based data sets and reanalyses. Northward of 60° S, the "Twentieth Century Reanalysis" for this period (20CRv3, Slivinski et al., 2019) fits best with their reconstruction whereas other products showed spurious trends. Poleward of 60° S the quality of all products deteriorates prior to 1957 due to the sparseness of pressure data. There is thus a need to improve reconstructions of Southern Hemisphere atmospheric circulation in the early 20[th] century.

Based on the assessment by Fogt and Connolly (2021), 20CRv3 is a good starting point for studying this period. However, although it is widely and successfully used, very little pressure data was ingested into 20CRv3 during these years. In fact, the ICOADS archive (Freeman et al., 2017) shows massive gaps in the South Atlantic for this period. In this paper, we present newly digitised ship log data and incorporate them into 20CRv3 in an offline assimilation approach (following Brönnimann, 2022). In addition to pressure from ships, we also assimilated one pressure series and five temperature series from land stations. A second data set on atmospheric circulation used in this study is the palaeoreanalysis ModE-RA (Valler et al., 2023ab), which combines model simulations and monthly-to-seasonal observations in an offline approach. Results based on these data sets are compared with purely observation-based data sets.

The paper is organised as follows. Section 2 presents the digitised ship logs and the data assimilation approach. In Section 3 we show the results from all data products. These are discussed in light of previous literature on the subject in Section 4. Conclusions are drawn in Section 5.

## 2    Data and method

### 2.1    Digitising of log books

We digitised the logs from 13 ships during the period 30 Apr. 1902 to 20 Sep. 1916 (Table 1). The ships were selected such as to give a good coverage of the Southern Ocean during the period of study. The tracks of the ships are shown in Fig. 1. Note that only one ship used the Panama canal, opened in 1914, giving a comparably good coverage of the Southern Ocean. In total we digitised 434'000 observations made at 64'080 observation times. All data were submitted to ICOADS.

For the application in this paper, we only used data in the Southern Hemisphere. Furthermore, of the 4-hourly pressure data we used only those within +/-2 hours of 12 UTC. Note that only pressure was



later assimilated; for temperature, a correction of the diurnal cycle would have been necessary, for
which we have too little information. Also, SST data from many of the ships were already in the
boundary conditions of 20CRv3.
This filtering restricts the number of observations. For the assimilation, we use 8063 measurements
made on 4209 days (1.92 measurement per day). The data cover the period 11 May 1902 to 25 Aug
1916. They are seasonally well distributed. In terms of the latitudinal distribution, 75% of the data are
between 30° and 60° S, 42% between 40° and 50° S.

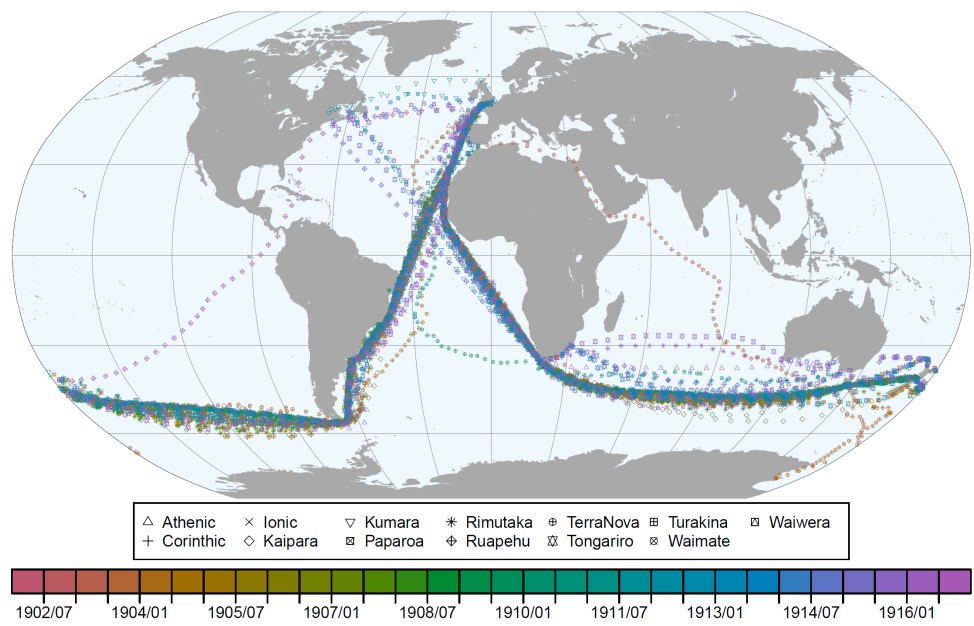


**Fig. 1.** Map of the tracks of the 13 ships for which data were digitised, coloured by year. Coordinates are shown
for 12 UTC, when they were measured.
**Table 1.** Periods covered by the ship logs, number of 12 UTC measurements assimilated, correlation with
20CRv3 and with 20CRv3+.

| Ship | period | n | $r_{20CRv3}$ | $r_{20CRv3+}$ |
|---|---|---|---|---|
| Athenic | 1905-1916 | 1208 | 0.78 | 0.98 |
| Corinthic | 1904-1907 | 312 | 0.84 | 0.86 |
| Ionic | 1904-1914 | 1349 | 0.81 | 0.96 |
| Kaipara | 1904-1908 | 157 | 0.87 | 0.97 |
| Kumara | 1911-1913 | 251 | 0.81 | 0.99 |
| Paparoa | 1905-1906 | 200 | 0.77 | 0.73 |
| Rimutaka | 1905-1916 | 1064 | 0.80 | 0.97 |
| Ruapehu | 1904-1916 | 316 | 0.71 | 0.94 |
| Terra Nova | 1903-1913 | 257 | 0.94 | 0.99 |
| Tongario | 1904-1916 | 1450 | 0.76 | 0.93 |
| Turakina | 1904-1915 | 431 | 0.81 | 0.99 |
| Waimate | 1902-1912 | 221 | 0.83 | 0.98 |
| Waiwera | 1913-1916 | 352 | 0.86 | 0.99 |




In addition, we also assimilated daily (12 UTC) series from land stations. This concerns one pressure
series that was not assimilated in 20CRv3 (Cape Town) as well as 5 temperature series from Uruguay,
South Africa, and Australia (see Table 2). The measurement hours did not change, hence, no further
adjustment of the diurnal cycle was necessary and the adjustment of the observations to 20CRv3, as is
described in the following, is sufficient. Note that further data would be available (e.g., pressure from
Buenos Aires), but is not yet digitised. The number of assimilated observations per year is shown in
Fig. 2.
**Table 2.** Land stations assimilated, latitude and longitude of the stations, variable (T = temperature, p =
pressure), number of 12 UTC measurements assimilated, correlation with 20CRv3 and with 20CRv3+ (note that
temperature data were deseasonalized).

| Station | Lat | Lon | var. | n | $r_{20CRv3}$ | $r_{20CRv3+}$ |
|---|---|---|---|---|---|---|
| Melbourne | -37.81 | 144.97 | T | 10955 | 0.76 | 0.96 |
| Kent Town | -34.92 | 138.62 | T | 10919 | 0.43 | 0.51 |
| Sydney | -33.86 | 151.21 | T | 10955 | 0.67 | 0.86 |
| Rocha | -34.49 | -54.31 | T | 9002 | 0.63 | 0.78 |
| Cape Town | -33.93 | 18.48 | T | 10224 | 0.59 | 0.89 |
| Cape Town | -33.93 | 18.48 | p | 10924 | 0.83 | 0.96 |


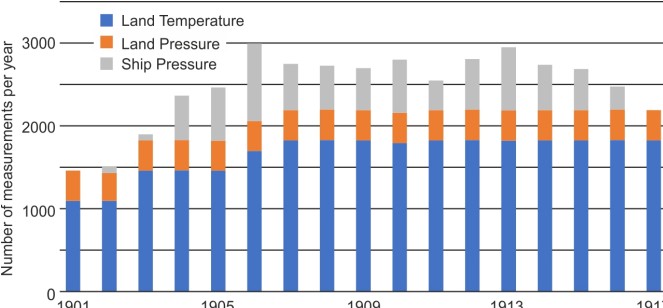


**Fig. 2.** Number of measurements per year in the period 1901-1917 (the numbers remain constant up to 1930 as
no further ship data were digitised and no land data are missing).

**2.2     The Twentieth Century Reanalysis**
The Twentieth Century Reanalysis (20CR) is a global dynamical reanalysis that is based on
assimilating surface pressure and sea-level pressure (SLP) data into an ensemble of atmospheric
model simulations (Compo et al., 2011). The current version 20CRv3 (Slivinski et al., 2019) starts in
1806 and comprises 80 members, with a spatial resolution is ca. 0.7° x 0.7°. It assimilates data from
the International Surface Pressure Databank (ISPD) Version 4.7 (Cram et al., 2015). We use 20CRv3
for our study and aim to assimilate additional observations. As we perform the assimilation off-line



(i.e., we do not cycle the analysis field back to the next model forecast step), the state vector does not
need to cover the full model state. For our analysis we use the fields of the Southern Hemisphere for
temperature and SLP at 12 UTC, from 1901 to 1930.
**2.3  Processing of observations**
All observations were first debiased relative to 20CRv3. For station temperature data we fitted the
first two harmonics of the seasonal cycle to both observations and 20CRv3 and subtracted the
difference. For pressure data (including the ship data), we corrected only the mean bias. We used the
overlap between observations and 20CRv3 in period 1901-1930 for debiasing (although the ship
records are much shorter).
For the assimilation of temperature, we assume an error of $3^2$ K$^2$, similar as in Brönnimann (2022), for
SLP $3^2$ hPa$^2$. Note that this concerns the difference between the data from the closest grid point
extracted from 20CRv3 and the observations. Thus, it accounts for the errors in the measurement
itself, the processing (Brugnara et al., 2015), and the representativity of the grid point but not the error
in 20CRv3 itself. Note that the debiasing step removes part of the systematic error.
**2.4  Offline data assimilation method**
The assimilation uses the Ensemble Square Root filter (Whitaker and Hamill, 2002) to assimilate
historical observations $\mathbf{y}$ into the 80 member ensemble of 20CRv3 ($\mathbf{x_b}$), yielding $\mathbf{x_a}$. For the following
see Brönnimann (2022).
First the ensemble mean is updated and then anomalies from the mean:
$$\overline{\mathbf{x_a}} = \overline{\mathbf{x_b}} + \mathbf{K}\left(\overline{\mathbf{y}} - \mathbf{H}\overline{\mathbf{x_b}}\right) \tag{1}$$
$$\mathbf{x'_a} = \mathbf{x'_b} + \widetilde{\mathbf{K}}(\mathbf{y'} - \mathbf{H}\mathbf{x'_b}) = \left(\mathbf{I} - \widetilde{\mathbf{K}}\mathbf{H}\right)\mathbf{x'_b}, \text{with}: \mathbf{y'} = 0 \tag{2}$$
$\mathbf{H}$ is the Jacobian matrix of the linear observation operator that extracts the observation from the model
state. The Kalman gain matrix $\mathbf{K}$ for the ensemble mean and the anomalies from the mean is defined
as:
$$\mathbf{K} = \mathbf{P^b}\mathbf{H^T}\left(\mathbf{H}\mathbf{P^b}\mathbf{H^T} + \mathbf{R}\right)^{-1} \tag{3}$$
$$\widetilde{\mathbf{K}} = \mathbf{P^b}\mathbf{H^T}\left[\left(\sqrt{\mathbf{H}\mathbf{P^b}\mathbf{H^T} + \mathbf{R}}\right)^{-1}\right]^{\mathbf{T}} \times \left(\sqrt{\mathbf{H}\mathbf{P^b}\mathbf{H^T} + \mathbf{R}} + \sqrt{\mathbf{R}}\right)^{-1} \tag{4}$$
$\mathbf{P^b}$ and $\mathbf{R}$ are the background error and observation error covariance matrices, respectively. The
former is calculated from the 80 members; no localisation was performed. The latter is assumed
diagonal. We did not store all 80 updated members, but only the ensemble mean and the ensemble
spread. The data set is published along with this paper in a repository (Brönnimann, 2023). The





assimilation was performed for the period 1901-1930. Outliers were removed **y–Hx** was larger than 3
$\times$ (**$HP^bH^T + R$**)$^{0.5}$.
The results were evaluated by means of the Pearson correlation and root mean squared error (RMSE)
at the observation locations (again, the mean annual temperature cycle was removed beforehand by
fitting the first two harmonics) as well as the reduction of the ensemble standard deviation. Note that a
full leave-one-out approach would be computationally expensive and of little value as measurements
are typically far apart from each other (results are expected to be similar as described in Brönnimann
(2022) for 1877/78, i.e., rather small but consistent improvements). We show results from a leave-
one-out approach only for a case when several observations were close together. As the assimilation is
offline and only few observations are available, the fields for each individual day only show
improvement in a few spotlight locations. Therefore, in the following, we aggregate the daily fields to
seasonal, annual or zonal means. At the same time, as all observations are debiased relative to
20CRv3, they will not have an effect on the long-term average.

**2.5     Other data sets**
For further analyses we used the temperature data sets HadCRUT5 (Morice et al., 2021), GISTEMP
(Lensson et al., 2019; GISTEMP Team, 2023) , NOAAGlobalTemp (Huang et al., 2020), and
Berkeley Earth BEST (Rohde and Housefather, 2020). We also analysed the original 20CRv3 output
as well as the monthly global climate reconstruction ModE-RA (Modern Era Reanalysis), which is
based on assimilating historical observations (including ship-based pressure observations) and proxies
into a 20 member ensemble of atmospheric model simulation (termed ModE-Sim) in a similar way as
in this paper (Valler et al., 2023ab). The data set is focusing on the period prior to 1890, and therefore
the input data is "frozen" at that year.
The model simulations ModE-Sim, which we also analysed, were driven by SSTs from HadISST
(Titchner and Rayner, 2014) volcanic and solar forcing (for details see Hand et al., 2023). Note that
these data sets (20CRv3, HadCRUT5, ModE-RA, ModE-Sim) use very similar SST data. Hence they
are not independent of each other. We therefore also analysed ModE-RAclim, which is the same as
ModE-RA except that it uses a random selection of 100 model years and ensemble members from
ModE-Sim as prior. Hence, this data set does not see the SST forcing. Note that ModE-RAclim and
ModE-Sim are mutually independent. For all these data sets we used the ensemble mean.
Finally, we also used the seasonal SLP reconstructions for Antarctica from Fogt and Connolly (2021),
which reach back to 1905. As recommended by the authors, we used the standard reconstruction for
Dec.-Feb. and the pseudo reconstruction for all other seasons. Note that these data were not
assimilated into ModE-RA.





**2.6    Analysis**
A preliminary analysis of Southern Hemisphere temperature from HadCRUT5 (Fig. 3, annual means)
shows that the coldest multiyear period in the record was from 1908-1911 (grey shaded). We
therefore focus on this four-year period in the following and analyse this period relative to the 1901-
1930 mean, which is also the period over which the assimilation was performed.

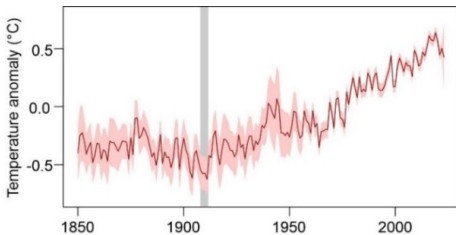


**Fig. 3.** Southern Hemisphere mean annual mean temperatures relative to the 1961-1990 reference period from
HadCRUT5 with 2.5% to 97.5% uncertainty range (grey).

This period is analysed first on the level of annual means in all available data sets. The standard
deviation of interannual variability is used to measure the magnitude of the anomalies. We then turn
to the seasonal scale and focus on atmospheric circulation as expressed in SLP.
**3    Results**
**3.1    Evaluation**
An analysis of correlations between 20CRv3 and the additionally digitised ship data shows that the
latter fit well with 20CRv3 (Table 1). One ship with a short record and many outliers exhibits a
correlation of only 0.71, the other ships show correlations between the 0.76 to 0.94. This points to the
already excellent quality of 20CRv3. Almost by construction, the assimilation approach greatly
improves the correlation to values of typically around 0.95 (the above-mentioned ship is an exception
to this).
The assimilation of additional data reduces the ensemble spread (Fig. 4, top). The spread reduction is
larger near land stations, where measurements are always available. A reduction is also seen in SLP
near the ship tracks, but since there were only few ships in the vast space, the reduction is statistically
weak, typically between 0.9 to 0.95 along the ship tracks.
The effect of the assimilation is illustrated for the example of 17 February 1909 in the lower part of
Fig. 4. On this day, five observations (two ship-based pressure measurements and three land-based
temperature measurements) were close to each other. For this case we performed a leave-one-out
approach. The raw 20CRv3 data show a high pressure system centred over Tasmania, which is further
strengthened and southward extended in 20CRv3+. In fact, the lower figure shows that both ships



indicated higher SLP than 20CRv3 (leftmost half-circle). Not surprisingly, in the leave-one-out
approach the pressure is increased at both locations due to the mutual effects of the two ships. The
departure from observations further increases in the full assimilation. Interestingly, the largest change
due to the assimilation does not occur exactly at the assimilation location, but to the south. Also, the
assimilation leads to a pressure decrease over some subpolar regions.

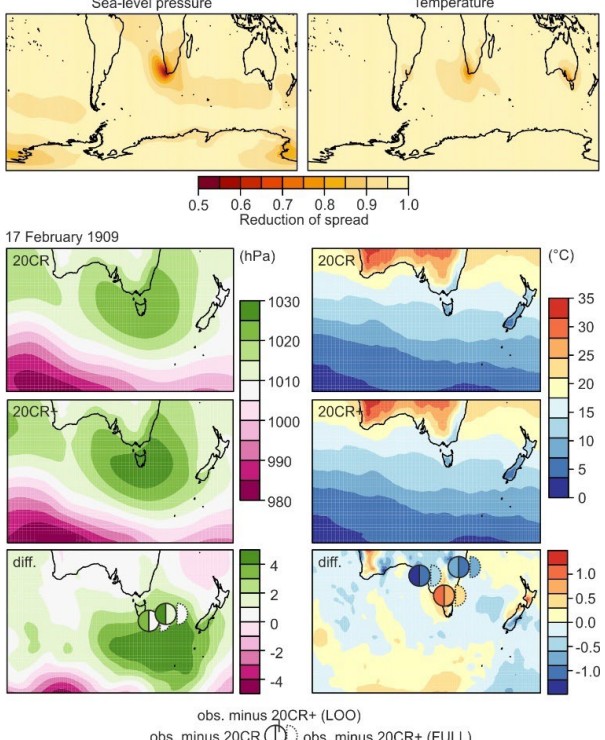

**Fig. 4.** Results of the evaluation of the assimilation approach. Top: Reduction of the average ensemble spread for SLP and temperature. Bottom: Results from the leave-one-out (LOO) approach for 17 February 1909 over Southern Australia.

For temperature, the situation is more complex as observations from Kent Town (Adelaide) and
Sydney were cooler than 20CR, whereas measurements from Melbourne were warmer. In Kent Town
and Melbourne, the leave-one-out approach reduces this departure. Hence, the stations mutually
correct each other in the right direction. An exception is Sydney, where the departure becomes
slightly larger when assimilating only all neighbouring stations. As for pressure, we note that the
change due to the assimilation is non-local and appears rather noisy with effects relatively far away
from observations. This is due to the imperfect estimation of the covariance matrix based on the 80
members. Localisation would help to remove this effect. On the other hand, the changes are typically
not very large.





## 3.2 Temperature

After having evaluated the assimilation, we turn to the analysis of the temperature. Annual mean
temperature anomaly maps are shown in Fig. 5 (top) for four available observation-based temperature
products. All show a general cooling, with particular cool spots in the southern tropical and
subtropical Atlantic, the tropical Pacific, and the South Pacific. Some land areas of the southern
midlatitudes were cold, too. Conversely, the ocean was warm around the Antarctic Peninsula.
However, the data basis is sparse and hence differences between different products considerable. The
middle and bottom part of the figure show results from assimilation approaches that incorporate
pressure and other variables. 20CRv3 shows a rather similar pattern as the observation-based data
sets. Over the ocean this is due to the fact that the model uses observed SSTs as boundary conditions,
but there is also agreement over land. For instance, as the observation-based products, 20CRv3
indicates low temperatures over Australia. The assimilation of additional data has only a very small
effect that is hardly visible when plotting only the anomaly field. Only when directly plotting the
difference one sees that the additionally assimilated information produces slightly warmer conditions
over Southern Australia.

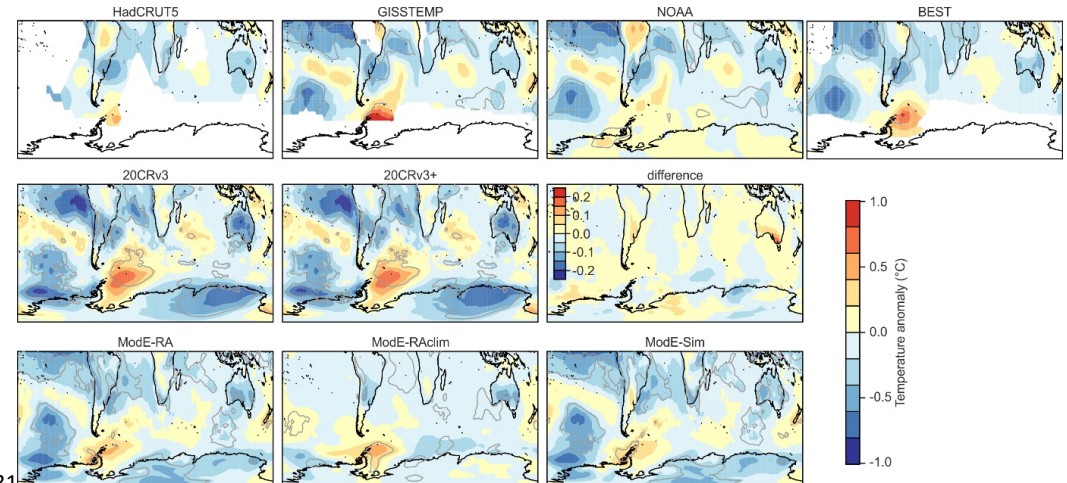

**Fig. 5.** Annual mean temperature difference between 1908-1911 and the remaining years in the 1901-1930
period in (top row) four observational data sets, middle row: 20CRv3 and 20CRv3+ as well as their difference,
and (bottom row) ModE-RA, ModE-RAclim, and ModE-Sim. Grey lines indicate where the 4-year anomaly
exceeds one standard deviation of the interannual variability of the annual mean in 1901-1930. The number of
missing values was not restricted.

While 20CRv3 only assimilated pressure data, ModE-RA (shown in the lower row) also assimilates
temperature; the resulting temperature anomaly map looks again very similar as that of all other
products. However, ModE-RA allows to disentangle where the information comes from. ModE-Sim
(lower right) indicates the pure atmospheric simulations, which again reproduce many of the



temperature features also over land. ModE-RAclim, in turn, only sees climatological SSTs and thus
shows the effect of only the observations. Interestingly, also ModE-RAclim shows a very similar
pattern. Note that ModE-RAclim and ModE-Sim are independent.
Time series of Southern Hemisphere temperature from these data sets (Fig. 6a) show a relatively good
agreement between the data sets over time. The period 1908-1911 stands out as the coldest 4-year
interval within the displayed period. The preceding drop as well as the small subsequent drop coincide
with volcanic eruptions. The good agreement between the data sets is partly due to similar SSTs used
in the different approaches. ModE-RAclim, which does not see SSTs or volcanic eruptions, shows a
weaker cooling for 1908-1911 (though still a cooling) but does show the cooling spikes before and
after.

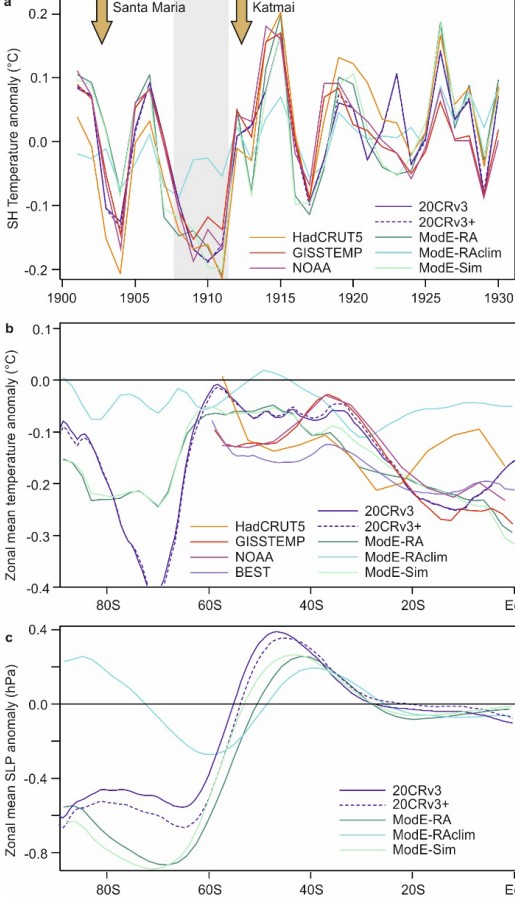


**Fig. 6.** Southern Hemisphere annual mean average temperature (anomaly from 1901-1930) in different data sets
(**a**). Zonal mean annual mean difference between 1908-1911 and the 1901-1930 climatology for temperature (**b**)
and SLP (**c**) in different data sets.



### 3.3    Pressure

The SLP anomaly field (Fig. 7) is again similar in 20CRv3 and in ModE-RA. In 20CRv3, the assimilation has a slightly larger impact on SLP than on temperature. Specifically, the assimilation leads to lower SLP in the Amundsen-Bellingshausen Sea area but also over the Southern Indian Ocean. ModE-RA again shows a good agreement between the pure simulation (which is arguably strongly affected by SSTs), the effect of only the observations, and the combined effect. Altogether, the fields indicate a positive SAM, although it should be noted that hardly any pressure data from the southern high latitudes enters any of the products.

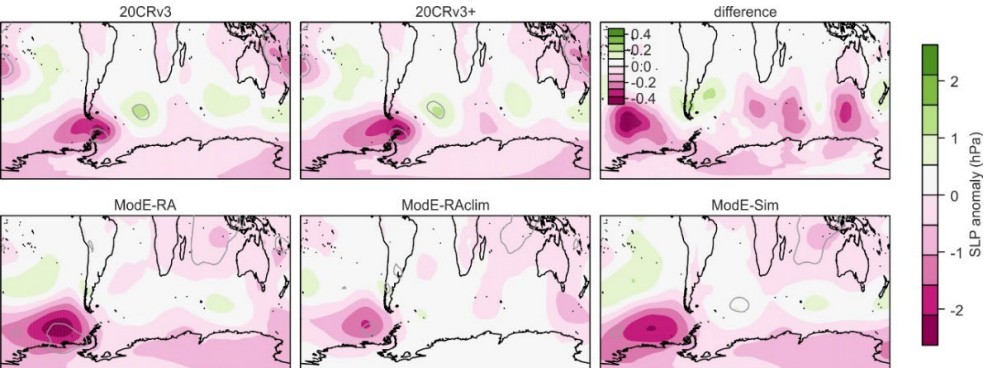

**Fig. 7.** Annual mean SLP difference between 1908-1911 and the remaining years in the 1901-1930 period in (top) 20CRv3 and 20CRv3+ as well as their difference, and (bottom) ModE-RA, ModE-RAclim, and ModE-Sim. Grey lines indicate where the 4-year anomaly exceeds one standard deviation of the interannual variability of the annual mean in 1901-1930.

In the next step we analysed zonally averaged temperature and SLP in 20CRv3 and 20CRv3+ (Fig. 6b,c) in order to obtain a better view of the possible SAM variability and of the changes induced by the assimilation to SLP and temperature. With respect to temperature, it becomes clear that the assimilation of additional observations (although most of them were pressure) led to a slightly smaller cooling at southern mid-to-high latitudes. However, the much larger signal is that in the tropics, which remains unaffected (note that only ships in the tropical Atlantic were assimilated).

The corresponding plot for SLP shows the sign of a positive SAM, with positive anomalies at mid latitudes and negative in the subpolar regions. This is slightly amplified in 20CRv3+. Interestingly, even ModE-RAclim, which does not see forcings, shows a very similar pattern as the other products north of 60° S. In the following we first focus on this change in the polar and subpolar circulation, then we address the circulation in the tropical region.

The strengthened SAM and specifically the strengthened Amundsen-Bellingshausen seas Low is further analysed on a seasonal scale by comparing 20CRv3+ and ModE-RA with the Fogt and Connolly (2021) data (Fig. 8). While all show a strengthening of the Amundsen-Bellingshausen seas



low, there are clear differences in the seasonal expression. ModE-RA has the strongest signal in fall,
20CRv3+ and Fogt and Connolly (2021) have the strongest signal in winter. Without further
information, it is impossible to rule out one or the other analysis, but it shows that there is still large
uncertainty with respect to Antarctic SLP despite the relatively good agreement on the annual mean
anomaly over this 4-year period.
Finally, we analysed the tropical atmospheric circulation. Figure 9 shows the SOI calculated in the
five data sets used in this study that contain SLP data. Generally, all data sets suggest a strengthened
Walker circulation around 1910, and the correlation between different data sets is relatively good.
Overall, however, the 1908-1911 period does not stand out as an extremely anomalous period in the
tropical Pacific.

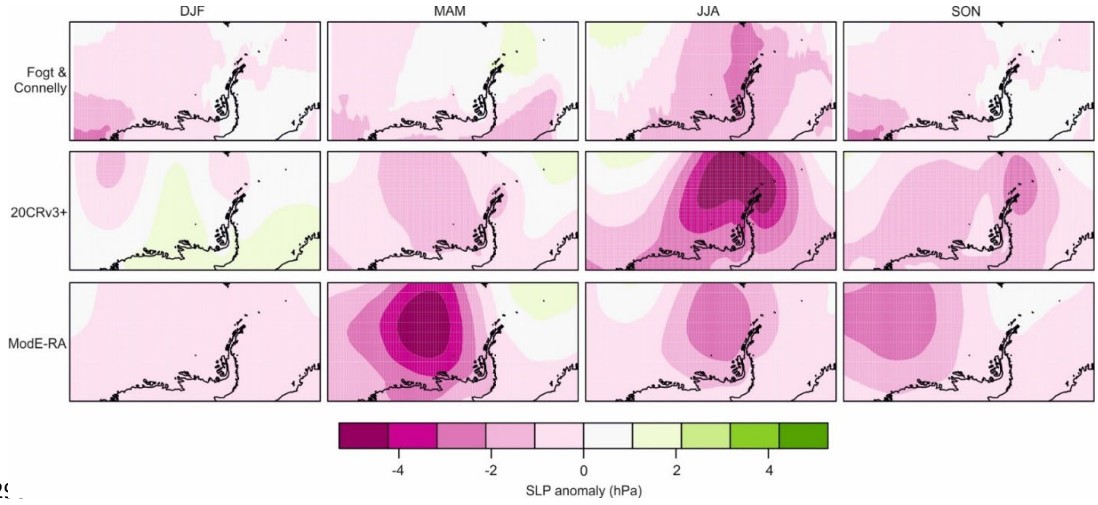

29

**Fig. 8.** Seasonal maps of SLP deviations in 1908-1911 from the 1901-1930 climatology for three different data
sets.

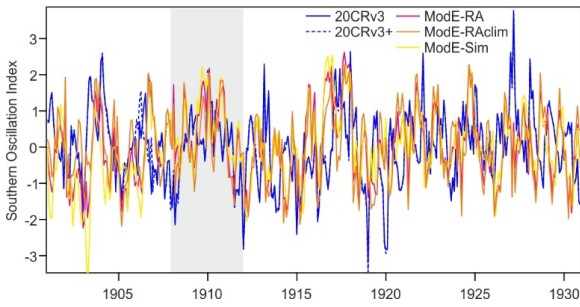


**Fig. 9.** Southern Oscillation Index calculated from 20CRv3, 20CRv3+, ModE-RA, ModE-RAclim, and ModE-
Sim calculated from the 1901-1930 climatology.





## 4    Discussion

The goal of this study was to generate improved data products to study the early 20th century cool period by digitising additional data and assimilating them into 20CRv3. A reduction of the spread could be achieved, and effects of the observations are clearly seen, although the overall effects on the 4-year averaged anomaly are small for temperature. At the same time, it should be noted that a large amount of observations would still be available and await to be rescued, and our study has shown that they could provide a large additional benefit.

The comparison of very different products such as 20CRv3, the different ModE-RA products, and the Fogt and Connolly (2021) SLP proved beneficial and also indicated a robustness of the signal, despite significant seasonal differences.

The analysis of different products for temperature and SLP for the years around 1910 in the Southern Hemisphere showed a good agreement of the spatial pattern relative to neighbouring years, at least on the annual mean scale. It should be noted, however, that a good agreement does not imply a small uncertainty as some of the data sets are based on very similar input. For instance, all use HadISST1.1 or similar for SSTs, and both ModE-RA and 20CRv3 use SLP (HadCRUT5 and ModE-RA both use land temperature data).

The assimilation of additional data into 20CRv3 leads to a slightly weaker cooling signal, but the magnitude of the difference is small. This might partly be due to the debiasing, which removes any signal at the scale of the length of the records. As most records (notably, the ship data) are short, their contribution to a 4-yr signal is necessarily weak. In all, the assimilation shows that the amplitude of the cooling is not well constrained with the additional data. Further data are needed to better understand the cooling phase.

A cooling phase nevertheless remains, and this cooling phase in the Southern midlatitudes around 1910 is a relevant period for better understanding natural decadal climate variability. The SST anomaly fields resembles a La Niña pattern (see also Fogt and Connolly, 2021), and the SOI indicates a strengthened Walker circulation around 1910. The general pattern of cooling as well as the pattern of SLP anomalies is very similar in all data sets, but overall the anomaly during these years does not seem to be extremely strong.

All data sets consistently show a neutral to positive SAM, in 20CRv3+ the positive SAM is even amplified. The SST pattern is generally consistent with a positive SAM (e.g., Hartmann, 2022, for a wind-based SAM index in Oct-Mar). The relation between La Niña conditions and a strong Amundsen sea low is also well known (Turner et al., 2013). Furthermore, the temperature anomaly pattern also resembles other suggested modes of variability such as the Subtropical Indian Ocean Dipole Mode or Southern Suptropical Atlantic Dipole Mode (e.g., Wainer et al., 2021; Yu et al., 2023), whose behaviour in the early 20th century is however not well known.



Finally, it should be noted that the first part of the 20th century also saw two large volcanic eruptions, namely Santa Maria in 1902 and Novarupta in 1912. The former was a tropical eruption that might have affected the Southern Hemisphere and had a global cooling effect; many ship logs from the Southern Ocean for these years have been imaged and await digitisation. The latter was a northern high-latitude eruption and might not have affected the Southern Hemisphere strongly, but the effect on Northern Hemisphere temperature is well studied (e.g., Oman et al., 2005). Two cooling spikes are seen in the Southern Hemisphere temperature series following these two eruptions, indicating a volcanic contribution to the cooling.

Taken together, the global cooling episode in the early 20th century, which peaks in 1908-1911, seems to be a combination of two volcanic eruptions, a strengthened Walker circulation around 1910, a positive SAM, and concurrent states in the South Atlantic and Indian Ocean Modes of variability. Although data uncertainty remains high, different data sets are consistent with each other and the patterns found are consistent with literature findings, thus supporting that the globally cool episode in the early 20th century was real. The ETCW, which was mostly studied with respect to Northern Hemispheric anomalies and the question of a warm phase in the 1940s (Brönnimann, 2009), emerged from a cold climate state strongest in the Southern Hemisphere.

## 5    Conclusions

A global cold period from 1908 to 1911 was analysed based on reanalysis data and observations. The cooling was most pronounced over the Southern Ocean, where available observations are few and far apart. Therefore, we digitized additional data from ship from the Southern Hemisphere from 1902-1916. These data, together with six land station records, were then assimilated offline into the 80-member ensemble of 20CRv3. The new data confirmed the temperature and pressure anomaly patterns found in 20CRv3. However, they decrease the ensemble spread, thus contributing to a smaller uncertainty of the analysis of atmospheric circulation. Overall, we find a positive SAM, which is even amplified in 20CRv3+. This is due to a strengthened Amundsen-Bellingshausen seas low, which is identified in all data sets. Differences between data sets emerge when analysing the seasonal timing of the anomalies.

SST and SLP patterns indicate a La Niña tendency and a strengthened Walker circulation around 1910, which is consistent with the strengthened Amundsen-Bellingshausen seas low. SST patterns in the South Atlantic and South Indian Ocean also are consistent with modes discussed in the literature. All of this results point to a real climatic phenomenon as the cause of the 1908-1911 cold anomaly and not a data artefact. Atmospheric model simulations using SSTs and external forcings as boundary conditions reproduce the main features of the SLP anomaly fields over the Southern Hemisphere found in a data set based only on observations. This again indicates that the temperature anomaly is physically consistent with all other information and perhaps an ocean-forced signal. Most importantly,

the period was preceded and followed by two volcanic eruptions, leading to global cooling. The
eruptions and the cold 1908-1911 period together provided a cold start into the ETCW.
Finally, the newly digitised ship log book data constitute only a small fraction of the non-digitised
(but imaged) log book data. Digitising the vast amount of marine data from the early 20th century
could help to generate an improved version of 20CR that would provide further insights into this cold
period.
**Funding:** This work was supported by the UK Newton Fund within the framework of the Weather and Climate
Science for Service Partnership (WCSSP) South Africa (WCSSP SA22_1.3), by the European Commission
(ERC Grant PALAEO-RA, 787574), by the Swiss National Science Foundation (188701) and by the NERC
project GloSAT (NE/S015647/1).
**Acknowledgements:** We would like to thank the students who digitised the ship logs for this work.
**Contributions:** YB coordinated the digitisation, processed, quality-controlled, and formatted the data. SB
performed the assimilation and the analyses, CW imaged and provided the log books. All authors commented on
the manuscript.
**Competing interests:** The authors declare no competing financial interests.
**Data availability statement:** The 20CRv3+ data set, including the station and ship input data and the R code
for the pre-processing and assimilation is available from Brönnimann (2023). The digitised ship log data have
been sent to ICOADS (Freeman et al., 2017). The 20CRv3 data (Slivinski et al., 2019) can be downloaded from
NCAR (https://rda.ucar.edu/datasets/ds131.3/, DOI: 10.5065/H93G-WS83, downloaded 7 Oct. 2022, last
accessed 11 Oct 2023). The ModE-RA data (Valler et al., 2023a) can be downloaded from DKRZ
(https://www.wdc-climate.de/ui/entry?acronym=ModE-RA). Antarctic SLP fields are available from Fogt and
Connolly (2021), HadCRUT5 is downloadable from https://www.metoffice.gov.uk/hadobs/hadcrut5/
(downloaded 14 Jul. 2023, last accessed 11 Oct. 2023). GISTEMP was obtained from the NASA Goddard
Institute for Space Studies (GISTEMP Team, 2023, downloaded 18 Jul. 2023, last accessed 11 Oct. 2023 at
https://data.giss.nasa.gov/gistemp/). The BEST data Rohde and Hausfather, 2020) were obtained from
https://berkeleyearth.org/data/ (downloaded 17 Jul. 2023, last accessed 11 Oct. 2023). NOAAGlobalTemp
version 5 (Huang et al., 2020) was downloaded from https://www.ncei.noaa.gov/products/land-based-
station/noaa-global-temp (downloaded 17 Jul. 2023, last accessed 11 Oc.t 2023).

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
