# Peer review of "Early Twentieth Century Southern Hemisphere Cooling"

_Climate of the Past, 2023_

## Author Comment (AC2)

**Reply to reviewer 2:**

1. Line 49: maybe clarify "very little pressure data was ingested into 20CRv3 during these years, **\*particularly in the Southern Hemisphere\***". In addition, the authors may wish to include a link to maps of the observations assimilated in 20CRv3 by year: https://psl.noaa.gov/data/20CRv3_ISPD_obscounts/

Thanks, we rephrase the sentence using the clarification and add a link to the ISPD visualization.

2. Line 51: maybe here is a good place to introduce 20CRv3+, since it's used in the Table caption prior to introduction in the text.

Thanks – the other reviewer commented the same. We now introduce 20CRv3+ already here.

3. Section 2.2: I recommend adding a few more details about 20CRv3 to this section. In particular, note that 20CRv3 does not use HadISST SSTs in the period of study, but rather SODAsi.3 (Giese et. al 2016, https://doi.org/10.1002/2016JC012079); there are also places later in the paper where it is stated that 20CRv3 uses SSTs "similar" to HadISST1.1 (lines 160 & 313). It might also be worth mentioning that 20CRv3 uses a cycling EnKF (unlike the offline approach used for 20CRv3+). Finally, it may be worth emphasizing that no station temperature data went into 20CRv3 at all; although the text in this paper is accurate, readers often miss this detail (in my experience.)

Thanks, we gladly add these clarifications.

4. Line 189/Figure 4 caption: Clarify whether this is reduction in time-averaged spread over some time period? If so, is there a reason for not (also) showing reduction in spread for the single date plotted in the lower part of the figure? Also, if this is time-averaged spread, then on line 191, you may want to add "since there were only few ships in the vast space **\*and time period\***, the reduction is statistically weak"

Yes, it is time-averaged spread. We will rephrase the sentence. We will also add the reduction in spread for the specific case in contour lines.

5. Line 225: Suggest changing "observed SSTs" to "prescribed SSTs", since 20CRv3 actually uses an ocean reanalysis for its SSTs in this period (see comment 3 above).

Yes, thanks, this is changed.

6. Line 307: clarify what "beneficial" means here, beyond the indication of a robust signal. Maybe, that the comparison also shows where uncertainty remains?

Yes, good point, we will at this statement.

Technical corrections

1. Line 102: "spatial resolution is" should be "spatial resolution of"

2. Line 124: maybe replace ybar with y (or define ybar)
3. Line 135: outliers were removed **\*when\*** y-Hx…
4. Line 186-187: Clarify that you're referring to 20CRv3+ here
5. Figure 4 caption: clarify what "results" are here (perhaps "analysis fields"?)
6. Line 226: "as the" should be "as with the"

Thanks, the corrections will all be incorporated.

---

## Author Response (AR1)

**Reply to reviewer 1**

Comments:

L39-46: The wording here is slightly misleading. The Fogt and Connollly (2021) dataset is a blend of the Fogt et al. (2019) spatial pressure reconstruction poleward of 60°S and the 20CRv3 equatorward of 60°S. Thus, everywhere north of 60°S the Fogt and Connolly merged data are exactly the 20CRv3. Poleward of 60°S, their data does agree best with 20CRv3 (it at least shows the least drift in the early 20$^{th}$ century). So perhaps the text could be revised to say, starting at line 42 "Poleward of 60°S, the "Twentieth century reanalysis" for this period fits best with their reconstruction, whereas other products showed spurious trends. However, there are marked differences between all products prior to 1957 south of 60°S due to the sparseness of pressure data...'. The Fogt and Connolly (2021) paper found more agreement in the various products over Southern Hemisphere landmasses in the early 20$^{th}$ century, for what it is worth.

Thanks, we will incorporate the sentence exactly as suggested.

L67: Grateful for you to submit these ship logbook data to ICOADS. Are there additional pressure data in recent releases of ISPDv4 that could also be used in this study, that aren't in ICOADS or assimilated into 20CRv3? Perhaps some early Antarctic data from British and Norwegian expeditions, as examples? Perhaps also the daily data (could maybe be close to the 12Z values) from Orcadas used in Zazulie et al. (2010), which from conversations with Gil Compo aren't included in the early 20$^{th}$ century in 20CRv3?

Thanks for the information. We were not aware of the Orcadas data. Since the project was computationally expensive, we cannot simply rerun the analysis, but we add a reference at the end of Section 2.1. In fact, we checked and Orcadas is not in ISPDv4.

"Also, a record from Orcadas station is available and not included in 20CRv3 (Zazulie et al., 2010)."

Table 1 – need to clarify what 20CRv3+ is in the caption – I assume it is the 20CRv3 with the additional data assimilated. If that is correct, you may wish to introduce the acronym when you first mention this dataset on L51-52.

Thanks, we add to L. 51-52:

"…in the following termed 20CRv3+".

L187-188: Technically, the above mentioned ship with a correlation to 20CRv3 of 0.71 (Ruapehu) does increase to 0.94 in 20CRv3+. It is the Paparoa ship with r=0.77 that actually decreases to r=0.73 in 20CRv3+.

Thanks for spotting this! We change the sentence to:

"One ship (Ruapehu) exhibits a correlation of only 0.71 (which then increases to 0.94 in 20CRv3+) … only for the "Paparoa", correlation decreases from 0.77 to 0.73"

Fig. 4: I'm having trouble interpreting the reduction of spread figure, perhaps overthinking it. The darker areas are the least reduction of spread based on the legend, which at face value to me would imply that they have had the least impact on the spread as they didn't reduce it much. But yet the text focuses on these as areas of improvement (near Cape Town, and along the ship tracks). Is the figure perhaps showing the ratio of spread, instead of the reduction in spread (which I take as the original spread (variance or standard deviation) minus the new spread (therefore, larger values are more of an improvement)? Please clarify in text or in caption.

You are absolutely right, "reduction of spread" is the wrong expression, it is the ratio of the spread in 20CRv3+ to the spread in 20CRv3. We now write in the caption of the figure:

"Ratio of the average ensemble spread ($\sigma_{20CRv3+}/\sigma_{20CRv3}$)"

And also change the figure accordingly.

Fig. 6a: can you comment in the text about what percentage the SH temperature anomaly is due to the tropical Pacific / La Niña -like cooling (Am I correct to assume this average is area-weighted by the cosine of latitude)? It seems like the zonal mean temperature plot shows much of the cold anomaly is in the tropics. I'm not sure much can be said of the temperature anomalies south of 60°S – what data constrain these over the Antarctic continent (the spread is also huge there!)? You can perhaps add a few sentences here after lines 272-273.

Yes, the temperature is area weighted, we now say this explicitly in the caption of Fig. 6 and add a sentence on the uncertainty south of 60° S.

Temperatures south of 60° S are largely unconstrained and should be treated with caution.

Fig 8 – note misspelling of Connolly in top row.

Thanks! We correct this.

Fig 8 – interesting comparison with the Fogt and Connolly (2021) data here – thanks for doing this! One important note here that is worth mentioning in the discussion following lines 283-284: The Fogt and Connolly (2021) data do include direct observations from Orcadas station, which starts in 1903. These data strongly control the solution of the Fogt and Connolly (2021) data near the Antarctic Peninsula, Weddell Sea / South Atlantic, in particular. Thus the positive pressure anomaly in MAM in Fogt and Connolly dataset near the tip of the Antarctic Peninsula is likely real (based on observations at Orcadas), whereas the strong positive pressure anomaly in 20CRv3+ over the northern Antarctic Peninsula may be overstated. You could quickly make comparisons of the pressure anomalies at the grid point of Orcadas station using the READER archive to verify these results – there is some information! (there are also data at Grytviken in South Georgia, if you want to further verify – these data are NOT used in Fogt and Connolly (2021).)

Thanks for the suggestion, we add a sentence on the interpretation and changed the final sentence of that paragraph. Now it reads:

The latter signal is presumably strongly influenced by the Orcadas station data (Zazulie et al., 2010). The comparison shows that there is still large uncertainty with respect to Antarctic SLP despite the relatively good agreement on the annual mean anomaly over this 4-year period.

Fig 9 – can you plot the observed SOI from the Australian BOM here as well?  http://www.bom.gov.au/climate/enso/soi/

Thanks, we added this data set to the figure. Moreover, we recalculated the indices as they were defined slightly differently (standardisation was done over all months and not by calendar month). We now explain the definition explicitly. The corresponding sentence reads:

We followed the definition of the Australian Bureau of Meteorology difference, i.e., the SLP difference between Darwin and Tahiti is standardised by calendar month and the result is multiplied by 10; we also show their index.

**Reply to reviewer 2**

General comments

This paper analyzes a cooling period in the Southern Hemisphere in the early 20[th] century using observations, existing reanalyses, and an augmented reanalysis in which additional observations were assimilated offline. I found the paper clear, logical, and well-organized. I just have some relatively minor comments.

 Specific comments

1. Line 49: maybe clarify "very little pressure data was ingested into 20CRv3 during these years, \***particularly in the Southern Hemisphere\***". In addition, the authors may wish to include a link to maps of the observations assimilated in 20CRv3 by year: https://psl.noaa.gov/data/20CRv3_ISPD_obscounts/

Thanks, we rephrase the sentence using the clarification and add a link to the ISPD visualization.

2. Line 51: maybe here is a good place to introduce 20CRv3+, since it's used in the Table caption prior to introduction in the text.

Thanks – the other reviewer commented the same. We now introduce 20CRv3+ already here.

3. Section 2.2: I recommend adding a few more details about 20CRv3 to this section. In particular, note that 20CRv3 does not use HadISST SSTs in the period of study, but rather SODAsi.3 (Giese et. al 2016, https://doi.org/10.1002/2016JC012079); there are also places later in the paper where it is stated that 20CRv3 uses SSTs "similar" to HadISST1.1 (lines 160 & 313). It might also be worth mentioning that 20CRv3 uses a cycling EnKF (unlike the offline approach used for 20CRv3+). Finally, it may be worth emphasizing that no station temperature data went into 20CRv3 at all; although the text in this paper is accurate, readers often miss this detail (in my experience.)

Thanks, we gladly add these clarifications:

It assimilates data from the International Surface Pressure Databank (ISPD) Version 4.7 (Cram et al., 2015) using a cycling Ensemble Kalman Filter. No station temperatures are assimilated. Note that 20CRv3 uses SSTs from the SODAsi.3 data set (Giese et al., 2016).

4.  Line 189/Figure 4 caption: Clarify whether this is reduction in time-averaged spread over some time period? If so, is there a reason for not (also) showing reduction in spread for the single date plotted in the lower part of the figure? Also, if this is time-averaged spread, then on line 191, you may want to add "since there were only few ships in the vast space **and time period**, the reduction is statistically weak"

Yes, it is time-averaged spread. We rephrased the sentence and we also added the reduction in spread for the specific case (red dashed and solid contour lines in the bottom plots).

[Figure]

**Fig. 4.** Results of the evaluation of the assimilation approach. Top: Ratio of the average ensemble spread ($\sigma_{20CRv3+}/\sigma_{20CRv3}$) for SLP and temperature. Bottom: Results from the leave-one-out (LOO) approach for 17 February 1909 over Southern Australia. Red curved in the bottom panels indicate the ratio of the average ensemble spread ($\sigma_{20CRv3+}/\sigma_{20CRv3}$) for 17 February 1909 (dashed = 0.8, solid = 0.5).

5.  Line 225: Suggest changing "observed SSTs" to "prescribed SSTs", since 20CRv3 actually uses an ocean reanalysis for its SSTs in this period (see comment 3 above).

Yes, thanks, this is changed.

6.  Line 307: clarify what "beneficial" means here, beyond the indication of a robust signal. Maybe, that the comparison also shows where uncertainty remains?

Yes, good point, we will at this statement.

Technical corrections

1. Line 102: "spatial resolution is" should be "spatial resolution of" Done
2. Line 124: maybe replace ybar with y (or define ybar) Done
3. Line 135: outliers were removed **\*when\*** y-Hx… Done
4. Line 186-187: Clarify that you're referring to 20CRv3+ here Done
5. Figure 4 caption: clarify what "results" are here (perhaps "analysis fields"?) Done
6. Line 226: "as the" should be "as with the" Done